# GLIDER: GLOBAL AND LOCAL INSTRUCTION-DRIVEN EXPERT ROUTER

## ABSTRACT

The availability of performant pre-trained models has led to a proliferation of fine-tuned expert models that are specialized to a particular domain or task. This has enabled the creation of powerful and adaptive routing-based "Model MoErging" (Yadav et al., 2024) methods with the goal of using expert modules to create an aggregate system with improved performance or generalization. However, existing MoErging methods often prioritize generalization to unseen tasks at the expense of performance on held-in tasks. This limitation adversely impacts practical applicability, as real-world deployments require robust performance across both known and novel tasks. We observe that current token-level routing mechanisms neglect the global semantic context of the input task. This token-wise independence hinders effective expert selection, particularly for held-in tasks, as routing decisions fail to incorporate the holistic semantic properties of the task. To address this, we propose a novel method, **G**lobal and **L**ocal **I**nstruction **D**riven Expert **R**outer (GLIDER) that integrates a multi-scale routing mechanism, encompassing a semantic global router and a learned local router. As recent LLMs demonstrate advanced reasoning capabilities for semantic-related contexts, the global router leverages this ability to enhance expert selection. By utilizing the input query and an LLM, the router generates semantic task instructions that guide the retrieval of the most relevant experts across all layers. This global guidance is complemented by a local router that facilitates token-level routing decisions within each module, enabling finer control and enhanced performance on unseen and challenging tasks. Our experiments using T5-based expert models for T0 and FLAN tasks demonstrate that GLIDER achieves substantially improved held-in performance while maintaining strong generalization on held-out tasks. Additionally, we perform ablations experiments to dive deeper into the components of GLIDER and plot routing distributions to show that GLIDER can effectively retrieve the correct expert for held-in tasks while also demonstrating compositional capabilities for held-out tasks. Our experiments highlight the importance of our multi-scale routing that leverages LLM-driven semantic reasoning for MoErging methods. Our code is attached as supplementary material.

## 1 INTRODUCTION

The emergence of highly capable large language models (LLMs) has marked an increased attention in downstream task specialization. This specialization often leverages parameter-efficient fine-tuning (PEFT) techniques, such as LoRA (Hu et al., 2021), which introduce minimal trainable parameters ("adapters") to adapt pre-trained LLMs for specific tasks. The compact size of these specialized PEFT modules enables easy sharing of these modules, which has led to the distribution of an evergrowing number of adapters on various platforms.

This proliferation of expert models, *i.e.* specialized adapters, has led to the development of methods for re-using such experts to improve performance or generalization (Muqeeth et al., 2024; Ostapenko et al., 2024; Huang et al., 2024a). Central to these approaches are routing mechanisms that adaptively select relevant experts for a particular task or query. These routing methods have been referred to as "Model MoErging" (Yadav et al., 2024) since they frequently share methodologies and ideas with mixture-of-experts (MoE) models (Shazeer et al., 2017; Fedus et al., 2022; Du et al., 2022) and model merging (Yadav et al., 2023b;a; Ilharco et al., 2022). However, MoE methods that train ex-

perts jointly from scratch (Gupta et al., 2022) while MoErging utilizes a decentralized, community-sourced pool of pre-trained experts. Furthermore, it departs from traditional model merging techniques by dynamically and adaptively combining these experts, optimizing performance at the query or task level. MoErging methods offer three key advantages: (1) They support decentralized model development by reusing and routing among independently trained experts, reducing reliance on centralized resources. (2) They facilitate modular capability expansion and "transparency" in updates as they either add or modify specialized expert models. 3) They allow for compositional generalization by recombining fine-grained skills from various experts, extending the system's abilities to new unseen tasks beyond the capabilities of the individual expert models.

Most existing methods for MoErging often prioritize performance on either known expert tasks (held-in) or generalization to unseen tasks (held-out) depending on their use cases (Chronopoulou et al., 2023; Muqeeth et al., 2024; Zhao et al., 2024). This specialization limits practical applicability, as real-world deployments demand robust performance across both held-in and held-out tasks. Consequently, existing methods exhibit suboptimal performance when evaluated on both held-in and held-out tasks, often leading to suboptimal overall performance. For example, while Phatgoose (Muqeeth et al., 2024) demonstrate strong performance on held-out data, they do not perform well on held-in tasks. We hypothesize that this gap arises from the model's token-level routing mechanism. We show that for the held-in tasks the independent routing decisions at each layer, based solely on individual token embeddings, lack sufficient global context to retrieve the correct expert for all token at every module. This leads to suboptimal routing which may propagate noise through the network, further hindering accurate expert utilization in deeper layers. This highlights a critical limitation of token-level approaches to handling both held-in tasks, which hence falls short of the goal of building a routing system that seamlessly handles arbitrary queries. We believe that adding a global routing mechanism based on semantic task information can further aid the token level router for correct retrieval for held-in tasks. Hence, we ask the question.

> *(Q) Can we leverage LLMs to generate semantics-aware task instructions to guide routing mechanism to facilitate both specialization and generalization?*

This paper addresses the challenges by investigating the potential of leveraging the inherent reasoning and generalization capabilities of LLMs to guide the routing process in an MoE-like model composed of specialized LoRA modules. We introduce, **G**lobal and **L**ocal **I**nstruction **D**riven **E**xpert **R**outer (GLIDER) that hinges on a multi-scale routing mechanism that contains both local and global routers as shown in Figure 2. The global router leverages LLM-generated, semantics-aware instructions (see Appendix A.2) to select the top-2 expert models for each input query across all the layers. This high-level guidance is then complemented by a learned local router, which makes token-level routing decisions at each module, enabling fine-grained control and improving performance on the challenging held-out tasks. Through this framework, we highlight the crucial role of LLM reasoning in unlocking the compositional generalization capabilities of MoE models.

To test the effectiveness of our GLIDER method, we follow Phatgoose (Muqeeth et al., 2024) and use T5 models (Raffel et al., 2020) to create expert models for T0 held-in (Sanh et al., 2022) and FLAN tasks (Longpre et al., 2023) and test performance on T0 held-in & held-out (Sanh et al., 2022) and big-bench lite (BIG-bench authors, 2023) & hard tasks (Suzgun et al., 2022). Our key contributions and findings are:

- We introduce GLIDER, which employs LLM-guided multi-scale global and local attention. Our experiments show that GLIDER outperforms previous methods, significantly improving performance on held-in tasks (*e.g.* 6.6% over Phatgoose on T0 held-in) while also enhancing zero-shot held-out compositional generalization (*e.g.* 0.9% over Phatgoose on T0 held-out).
- We find that without LLM assistance, MoE models underperform individual specialized models on held-in tasks by 8.2%. Incorporating semantic-aware instructions enables GLIDER to achieve comparable performance, demonstrating the LLM's capacity to effectively infer task identity and guide module selection without explicit task labels.
- GLIDER also maintains strong performance on held-out tasks, showcasing its adaptability and generalization capabilities. Our work highlights the critical role of LLMs in enhancing MoE models' compositional generalization, advancing the development of more robust and versatile AI systems capable of handling both familiar and novel tasks.

## 2 RELATED WORKS

**MoErging Methods.** The abundance of specialized expert models has spurred the development of techniques to leverage "experts" models for enhanced performance and generalization. Yadav et al. (2024) in their recent survey called such techniques as "MoErging" [1] methods which rely on adaptive routing mechanisms to select relevant experts for specific tasks or queries. These methods can be broadly classified into four categories based on the design of their routing mechanisms.

`Embedding − Based Routing` : This category encompasses methods that derive routing decisions from learned embeddings of expert training data. These methods typically compare a query embedding against the learned expert embeddings to determine the optimal routing path. Examples include AdapterSoup (Chronopoulou et al., 2023), Retrieval of Experts (Jang et al., 2023), Token-Level Adaptation (Belofsky, 2023), LoraRetriever (Zhao et al., 2024), Mo'LoRA (Maxine, 2023), the embedding-based approach of Airoboros (Durbin, 2024), and Dynamic Adapter Merging (Cheng et al., 2024).

`Classifier − Based Routing` : This category consists of methods that train a router to function as a classifier. This router is trained to predict the optimal routing path based on features extracted from expert datasets or unseen data. Representative methods in this category include Zooter (Lu et al., 2023), Branch-Train-Mix (Sukhbaatar et al., 2024), Routing with Benchmark Datasets (Shnitzer et al., 2023), Routoo (Mohammadshahi et al., 2024), and RouteLLM (Ong et al., 2024). The key distinction between embedding-based and classifier-based routing lies in the router's architecture and training methodology. While embedding-based routing often employs a nearest neighbor approach, classifier-based routing typically relies on logistic regression or analogous classification techniques.

`Task − Specific Routing` : This category focuses on methods tailored to enhance performance on specific target tasks. These methods learn a task-specific routing distribution over the target dataset to optimize performance for the given task. Methods in this category include LoraHub (Huang et al., 2023), LoRA-Flow (Wang et al., 2024), AdapterFusion (Pfeiffer et al., 2021), $\pi$-Tuning (Wu et al., 2023), Co-LLM (Shen et al., 2024), Weight-Ensembling MoE (Tang et al., 2024), MoLE (Wu et al., 2024), MeteoRA (Xu et al., 2024), PEMT (Lin et al., 2024), MixDA (Diao et al., 2023), and Twin-Merging (Lu et al., 2024).

`Routerless Methods` : This final category encompasses methods that do not rely on an explicitly trained router. Instead, these methods often employ alternative mechanisms, such as heuristics or rule-based systems, for routing decisions. Examples include Arrow ↗ (Ostapenko et al., 2024), PHATGOOSE (Muqeeth et al., 2024), the "ask an LLM" routing of Airoboros (Durbin, 2024) and LlamaIndex (Liu, 2024).

**Model Merging.** Model merging (Yadav et al., 2023b; Choshen et al., 2022; Wortsman et al., 2022; Ramé et al., 2022; Matena & Raffel, 2022; Ilharco et al., 2022; Tam et al., 2023; Jin et al., 2022; Yang et al., 2023) consolidates multiple independently trained models with identical architectures into a unified model that preserves individual model capabilities. While simple parameter averaging suffices for models within a linearly connected low-loss parameter space (McMahan et al., 2017; Stich, 2018; Frankle et al., 2020; Wortsman et al., 2021), more sophisticated techniques are necessary for complex scenarios. For instance, task vectors facilitate merging expert models trained on diverse domains (Ilharco et al., 2022). Additionally, methods like weighted merging using Fisher Importance Matrices (Matena & Raffel, 2022; Tam et al., 2023) and TIES-Merging, which addresses sign disagreements and redundancy (Yadav et al., 2023b) offers improved performance. As a non-adaptive expert aggregation method, merging serves as a fundamental baseline for numerous Model Editing with Regularization (MoErging) techniques.

**Multitask Learning (MTL).** research offers valuable insights for decentralized development. Notably, investigations into task-relatedness (Standley et al., 2020; Bingel & Søgaard, 2017; Achille et al., 2019; Vu et al., 2020; Zamir et al., 2018; Mou et al., 2016) provide guidance for designing routing mechanisms, while MTL architectures addressing the balance between shared and task-specific knowledge (Misra et al., 2016; Ruder et al., 2017; Meyerson & Miikkulainen, 2017; Zaremoodi

---

[1]See e.g. `https://huggingface.co/spaces/open-llm-leaderboard/open_llm_leaderboard`

et al., 2018; Sun et al., 2019) offer strategies for combining expert contributions in a decentralized manner.

**MoE for Multitask Learning.** Recent research has extensively investigated mixture-of-experts (MoE) models for multitask learning, achieving promising results in unseen task generalization. These approaches generally fall into two categories: (1) Example Routing: Studies like Muqeeth et al. (2023); Zadouri et al. (2023); Wang et al. (2022a) train routers to dynamically select experts for each input, while Caccia et al. (2023) demonstrate the efficacy of routing at a finer granularity by splitting expert parameters into blocks. (2) Task Routing: Ponti et al. (2023) employs a trainable skill matrix to assign tasks to specific parameter-efficient modules, while Gupta et al. (2022) leverages task-specific routers selected based on domain knowledge. Ye et al. (2022) proposes a layer-wise expert selection mechanism informed by task representations derived from input embeddings. Such approaches leverage task-specific representation to allow the router to effectively select the most suitable experts for unseen tasks. While these studies differ from our setting by assuming simultaneous data access, they offer valuable insights applicable to our exploration of creating routing mechanisms over expert models.

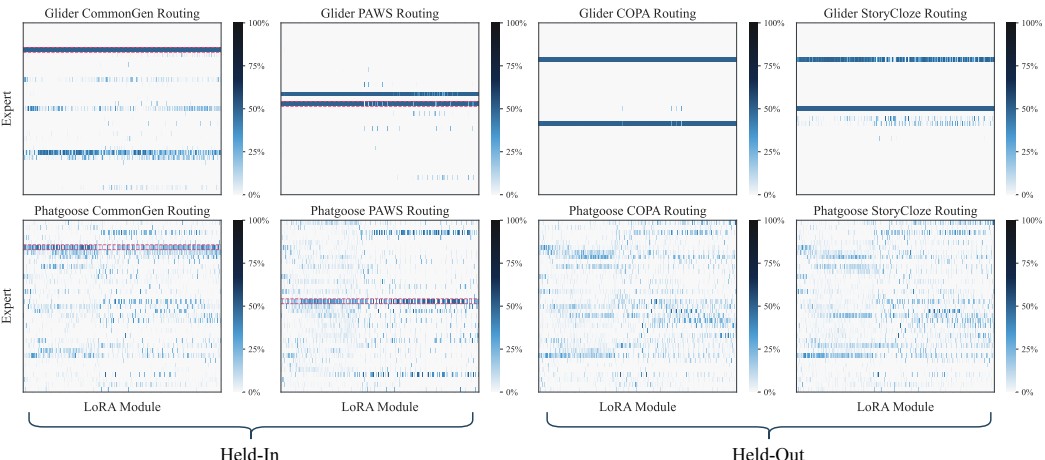

Figure 1: We present routing heatmaps for `GLIDER` and Phatgoose on two held-in and two held-out tasks. For held-in tasks, oracle experts are marked with red dashed lines. `GLIDER` selects oracle experts more frequently than Phatgoose for held-in tasks, leading to improvements of 3.3% on CommonGen and 6.5% on PAWS. For held-out tasks, `GLIDER` also tends to select the most relevant experts across most LoRA modules, resulting in improvements of 2.2% on COPA and 5.8% on StoryCloze.

## 3 PROBLEM STATEMENT

In our work, we aim to build a routing mechanism capable of performing well on diverse queries from various tasks, including both seen and unseen tasks. For each query/token and module, this routing mechanism dynamically selects a model from a large pool of specialized expert models to achieve high performance. To facilitate modular development, we adopt a *contributor-aggregator* framework (Yadav et al., 2024) where individual contributors create specialized expert models from a generalist model for their respective tasks and distribute these models to others for public usage. The aggregator builds a routing mechanism over the expert models that shared by the contributor to direct queries to the most relevant experts. Following recent works (Muqeeth et al., 2024; Ostapenko et al., 2024), we use parameter-efficient finetuning (PEFT) (Liu et al., 2022; Sung et al., 2022; Poth et al., 2023) methods like LoRA (Hu et al., 2022) to train the expert models. Since PEFT typically has lower computational and communication costs than full-model finetuning (Hu et al., 2022; Liu et al., 2022), the use of PEFT makes it easier to participate and contribute. PEFT methods introduce

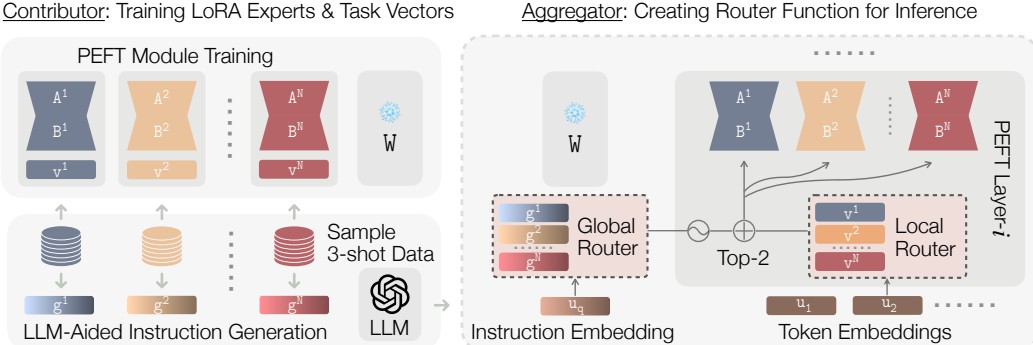

Figure 2: Overview of our method. Contributor (left): Each contributor utilizes local data to train several components: the PEFT module (comprising $\text{A}_i$ and $\text{B}_i$), task vectors ($\text{v}_i$), and global routing vectors ($\text{g}_i$). For the latter, an LLM is employed to generate semantically-informed instructions based on 3 randomly selected examples, which are then embedded into $\text{g}_i$. Aggregator (right): The aggregator utilizes local and global task vectors to construct local routers $[\bar{\text{v}}^1; \ldots; \bar{\text{v}}^N]$ and a global router $[\text{g}^1; \ldots; \text{g}^N]$, respectively. For each query, the global router uses an LLM-generated instruction embedding to produce the global routing score. This score is then scaled and combined with the local routing score, enabling fine-grained control over expert selection.

modules throughout the model – for example, LoRA (Hu et al., 2022) introduces a low-rank update at every linear layer in the model. We refer to each of these updates as a *module*. Subsequently, the trained expert models and additional information are shared with the aggregators. The aggregator's job is to collect these expert models and the additional information and design the post-hoc routing mechanism. This mechanism will effectively direct incoming queries to the most appropriate expert model for each token and at each module to ensure optimal performance on both seen and unseen tasks. This approach allows for the seamless integration of new capabilities by adding expert models to the existing pool. Next, we formally define our contributor-aggregator framework.

Let us assume that there are $N$ contributors, $\{c_1, c_2, \ldots, c_N\}$, and each contributor $c_i$ has access to a task-specific datasets $\mathcal{D}_i$. Each contributor, $c_i$, follows the predefined training protocol $\mathcal{T}$ provided by the aggregator. The training protocol ($\mathcal{T}$) takes in a base model ($\boldsymbol{\theta}_{\text{base}}$) and a dataset ($\mathcal{D}_i$). It returns the expert model parameters ($\phi_i$) along with any additional information ($\Psi_i$) that needs to be shared with the aggregators, for example, the gate vectors described in Section 4.1. Specifically, $\{\phi_i, \Psi_i\} \leftarrow \mathcal{T}(\boldsymbol{\theta}_{\text{base}}, \mathcal{D}_i)$. All contributors share this information with the aggregator, which creates a pool of models containing $\{(\phi_i, \Psi_i)\}_{i=1}^N$. The aggregators ($\mathcal{A}$) then uses these expert models and the auxiliary information to create a routing mechanism $\mathcal{R}(.)$ that takes the user query $q$ as the input and return routing path describing how the information is routed through the given set of expert models. Formally, $\mathcal{R}(.) \leftarrow \mathcal{A}(\{(\phi_i, \Psi_i)\}_{i=1}^N)$. The function $\mathcal{R}(.)$ describe the full path of input query by making various choices about 1) expert input granularity, choosing to route per-token, per-query, or per-task, 2) expert depth granularity, opting for either per-module or model-level routing, and 3) selecting between sparse or dense routing. Finally, the aggregator uses the routing mechanism to answer incoming queries.

## 4 METHODOLOGY

To recap, our goal is to build a MoErging method that dynamically routing queries to a diverse pool of specialized expert models, addressing the challenge of effectively handling queries from various tasks and ensuring both held-in and held-out performance. Our proposed method, **G**lobal and **L**ocal **I**nstruction **D**riven **E**xpert **R**outer (GLIDER), leverages a combination of local and global routing vectors to achieve this goal. Specifically, contributors train task-specific routing vectors, while a large language model (LLM) generates a global semantic task instructions which are then converted to global instruction routing vectors. During inference, these local and global routing vectors are

combined to perform top-k discrete routing, directing queries to the most suitable expert model. This process is visualized in Figure 2 and described in detail below.

### 4.1 Expert Training Protocol

Our expert training protocol $\mathcal{T}$ takes as input the base model parameters, $\theta_{\text{base}}$, and a dataset $\mathtt{d}$ and performs three steps to obtain the required output. First, we train the LoRA experts ($\phi$), then train the local routing vectors ($\mathtt{l}$) while keeping the LoRA experts fixed. Finally, we train obtain the global routing vector ($\mathtt{g}$) by using an LLM and an embedding model. Formally, in our case, $\phi$, $\Psi = \{\mathtt{l}, \mathtt{g}\} \leftarrow \mathcal{T}(\theta_{\text{base}}, \mathtt{d})$ which are then shared with the aggregators to create the routing mechanism. We described these steps in detail below.

**PEFT Training of Expert Model.** GLIDER is compatible with expert models trained using parameter-efficient finetuning methods (*e.g.* LoRA (Hu et al., 2022), Adapters (Houlsby et al., 2019)) that introduce small trainable modules throughout the model. We focus on PEFT experts because they typically have lower computational and communication costs than full-model fine-tuning (Yadav et al., 2023a), making it easier to train and share expert models. Following Phatgoose (Muqeeth et al., 2024), this work specifically focuses in LoRA (Hu et al., 2022) due to its widespread use. LoRA introduces a *module* comprising the trainable matrices $\mathtt{B} \in \mathbb{R}^{\mathtt{d} \times \mathtt{r}}$ and $\mathtt{A} \in \mathbb{R}^{\mathtt{r} \times \mathtt{n}}$ in parallel to each linear layer with parameters $\mathtt{W} \in \mathbb{R}^{\mathtt{d} \times \mathtt{n}}$. Given the $\mathtt{t}^{\text{th}}$ input token activation $\mathtt{u_i}$, LoRA modifies the output of the linear layer from $\mathtt{Wu_i}$ to $\mathtt{Wu_i} + \frac{\alpha}{\mathtt{r}} * \mathtt{BAu_i}$ where $\alpha$ is a constant and usually is set to 1. During training, the matrices $\mathtt{A}$ and $\mathtt{B}$ are trainable while the original linear layer $\mathtt{W}$ is kept frozen. We denote the final trained expert parameters with $\phi = \{(\mathtt{A_1}, \mathtt{B_1}), \ldots, (\mathtt{A_m}, \mathtt{B_m})\}$, where $\mathtt{m}$ is the number of modules in the model.

**Training Local Routing Vectors.** Following Phatgoose (Muqeeth et al., 2024), after training the PEFT modules on their dataset, a local router is introduced before each PEFT module. This router, employing a shared vector across all queries and tokens, dynamically determines the utilization of the PEFT module based on the input token activations. The router is trained for a small number of steps using the same dataset and objective as the PEFT module, while keeping the expert PEFT parameters fixed. This process effectively learns to associate the token activation patterns with the learned expert model. For LoRA, the local router, represented by a trainable vector $\mathtt{v} \in \mathbb{R}^{\mathtt{d}}$, controls the contribution of the PEFT module to the final output. This results in a modified linear layer of the form $\mathtt{Wu_i} + \frac{\alpha}{\mathtt{r}} * \mathtt{BAu_i} * \sigma(\mathtt{v}^{\top} \mathtt{u_i})$, where $\alpha$, $\mathtt{W}$, $\mathtt{B}$, and $\mathtt{A}$ are frozen, and the local router $\mathtt{v}$ is learned. We denote the final local routing vectors as $\mathtt{l} = \{\mathtt{v_1}, \ldots, \mathtt{v_m}\}$ where $\mathtt{m}$ is the number of modules in the model.

**Creating LLM-Aided Global Routing Vector.** The local routing vectors capture the intricate relationships between token activations and expert models, enabling efficient query routing in cases where no dedicated expert is available. Conversely, for queries corresponding to held-in tasks, direct retrieval of the relevant expert model is preferred to process the full query. For this purpose, we create a global routing vector that utilizes an LLM to generate a semantically-informed instruction, termed as task description, which effectively captures the essence of the kind of queries the expert can handle. We prompt an LLM with three randomly selected in-context examples to generate this task description. We used the `gpt-4-turbo` model along with the prompt provided in Appendix A. The resulting task description is then embedded using an off-the-shelf embedding model, specifically the `nomic-embed-text-v1.5` model, to produce a global routing vector for the task. We denote the global routing vector as $\mathtt{g} \in \mathbb{R}^{\mathtt{d_g}}$.

### 4.2 GLIDER: Inference Expert Aggregation Phase

Following training, all contributors share their expert models along with the auxiliary information comprising of the local and global routing vectors, $\{\phi^{\mathtt{t}}, \mathtt{l}^{\mathtt{t}}, \mathtt{g}^{\mathtt{t}}\}_{\mathtt{t}=1}^{\mathtt{N}}$ with the aggregators. The GLIDER method subsequently leverages this information to perform inference on arbitrary queries.

**Local Router.** Before each input module $\mathtt{m}$, a separate local router $\mathtt{L_m} \in \mathbb{R}^{\mathtt{N} \times \mathtt{d}}$ is inserted to make local per-token, per-module routing decisions. For a given module $\mathtt{m}$ and expert model $\mathtt{c}$, we first standardize the task-specific local routing vectors $\mathtt{v_m^c}$ by subtracting its mean and dividing by the

standard deviation to obtain $\bar{v}_m^c$. Next, we obtain the local router for module m by stacking these standardised local routing vectors as $L_m = [\bar{v}_m^1; \ldots; \bar{v}_m^N] \in \mathbb{R}^{N \times d}$. Next, for each token i with activation $u_i$ coming into module m, we standardise it to obtain $\bar{u}_i$. We then compute the local affinity scores, $s_m^{loc} \in \mathbb{R}^N$ between the local router $L_m$ and $u_i$ as $s_m^{loc} = \cos\text{-}\sin(L_m, u_i)$.

**Global Router.** The global router aims to capture task semantics to retrieve relevant experts for any given input query. We create the global router $G \in \mathbb{R}^{N \times d_g}$ by stacking the global routing vectors from all the expert models as $G = [g^1; \ldots; g^N]$. This router is not a part of the base model and is added before the model to independently process the fully query. Given an input query u along with three few-shot input-output pairs of similar queries, we prompt an LLM (`gpt-4-turbo`) using the template provided in Appendix A to obtain a task description for the query. We then embed this task description using the same embedding model (`nomic-embed-text-v1.5`) to obtain the vector $q_u \in \mathbb{R}^{d_g}$. We then compute the global affinity score, $s^{glob} \in \mathbb{R}^N$, by computing the cosine similarity as $s^{glob} = \cos\text{-}\sin(G, q_u)$.

**Combining Global and Local Router.** At each module m, we have the global and local affinity scores $s^{glob}$ and $s_m^{loc}$ respectively. Following Phatgoose (Muqeeth et al., 2024), we scale the local scores with a factor of $1/\sqrt{N}$. However, the global router's main goal is to retrieve the correct expert for the held-in tasks. Therefore, we first check if the expert with the highest global affinity score ($\max(s^{glob})$) is above a threshold (p). If such experts exist, then we set a high $\alpha$ to enforce retrieval and vice versa. Hence, we propose to scale the global scores with $\alpha$, where $\alpha = \gamma * \mathbb{I}_{\{\max(s^{glob}) - p > 0\}} + \beta$, where p is the cosine similarity threshold, and $\gamma$ and $\beta$ are scaling hyperparameters. Using our ablation experiments in Section 5.4, we set p = 0.8, $\gamma = 100$ and $\beta = 3$. We then obtain the final affinity score $s \in \mathbb{R}^N = \alpha * s^{glob} + s_m^{loc}/\sqrt{N}$. Then GLIDER selects the top-k experts after performing `softmax` over the final affinity score s as $\mathcal{E}_{top} = \text{top-k}(\text{softmax}(s))$. Finally, the output of the module for token activation $u_i$ is computed as $W u_i + \sum_{k \in \mathcal{E}_{top}} w_k * B_k A_k u_i$.

## 5 EXPERIMENTS

### 5.1 SETTING

**Dataset.** Our experiments utilize the multitask prompted training setup introduced by Sanh et al. (2021), which has become a standard benchmark for evaluating generalization to unseen tasks (Chung et al., 2022; Longpre et al., 2023; Jang et al., 2023; Zhou et al., 2022). Following Phatgoose (Muqeeth et al., 2024), we employ LM-adapted T5.1.1 XL (Lester et al., 2021) as our base model which is a 3B parameter variant of T5 (Raffel et al., 2020) further trained on the C4 dataset using a standard language modeling objective. For held-out evaluations, we follow Phatgoose (Muqeeth et al., 2024) and use three held-out benchmark collections. We use the T0 held-out (T0HO) datasets used in Sanh et al. (2021) and the two subsets of BIG-bench (BIG-bench authors, 2023). Specifically, we use BIG-bench Hard (BBH) (Suzgun et al., 2022), consisting of 23 challenging datasets, and BIG-bench Lite (BBL) (BIG-bench authors, 2023), a lightweight 24-dataset proxy for the full benchmark. Similar to Muqeeth et al. (2024), we exclude certain BIG-bench datasets due to tokenization incompatibility with the T5 tokenizer.

**Expert Creation.** To create the pool of expert module for routing, we follow Muqeeth et al. (2024) and use two distinct dataset collections: ❶ T0 Held-In (Sanh et al., 2021) consisting of the 36 held-in prompted datasets for tasks from the T0 training procedure. ❷ The "FLAN Collection" (Longpre et al., 2023) which significantly expands the T0 tasks by incorporating prompted datasets from SuperGLUE (Wang et al., 2019), Super Natural Instructions (Wang et al., 2022b), dialogue datasets, and Chain-of-Thought datasets (Wei et al., 2022b). Following Muqeeth et al. (2024), we create 166 specialized models from the FLAN Collection. For each dataset in these collections, we train Low-Rank Adapters (LoRAs) (Hu et al., 2021) modules resulting in pools of 36 and 166 expert models for T0 Held-In and FLAN, respectively. Similar to Phatgoose, we use a rank of $r = 16$ and train for 1000 steps using the AdamW optimizer (Loshchilov & Hutter, 2017) with a learning rate of $5 \times 10^{-3}$ and a warmup ratio of 0.06. After training the LoRA module, we freeze it and train the local routing vectors for an additional 100 steps with the same hyperparameters. Finally, following prior work (Shazeer et al., 2016; Du et al., 2022; Lepikhin et al., 2020), GLIDER performs top-$k$ routing with $k = 2$.

## 5.2 BASELINES

**Expert Merging.** Model Merging (Yadav et al., 2023b; Choshen et al., 2022) involves averaging the parameters of multiple models or modules to create a single aggregate model. We merge by multiplying the LoRA matrices and then taking an unweighted average of all the experts within the pool. It is important to note that this merging strategy requires homogeneous expert module architectures; in contrast, GLIDER can accommodate heterogeneous expert modules.

**Arrow.** Following Ostapenko et al. (2024), we employ a routing mechanism where gating vectors are derived from LoRA expert modules. Specifically, the first right singular vector of the outer product of each module's LoRA update ($BA$) serves as its gating vector. Input routing is determined by a probability distribution based on the absolute dot product between the input representation and each gating vector. We utilize top-$k$ routing with $k = 2$.

**Phatgoose.** Phatgoose (Muqeeth et al., 2024) first learn the LoRA modules for each, followed by learning a sigmoid gating vector similar to our local router. During inference, they make routing decisions for each token independently for all modules. Specifically, they first standardize the input token activations and gating vectors from all experts and then perform similarity-based top-2 routing.

**LoRA Hub.** LoraHub (Huang et al., 2023) method performs gradient-free optimization using few-shot task samples to learn mixing coefficients for different expert models while keeping them fixed. Once the coefficients are learned, they merge the experts with the learned weight and route through the merged expert.

**Multi-task Fine-Tuning.** While multitask training, a proven method for enhancing zero-shot generalization (Sanh et al., 2021; Wei et al., 2022a), is infeasible given our problem setting and data access limitations, we include it as a baseline using publicly available models. Specifically, we utilize the T0-3B model (Sanh et al., 2021) for the T0 Held-In datasets, given its training on a matching dataset collection. For FLAN, a directly comparable publicly available model is unavailable; therefore, we report results for FLAN-T5 XL, trained on a different, undisclosed dataset mixture, while acknowledging the limitations of this indirect comparison.

**Oracle.** Following Jang et al. (2023) and Muqeeth et al. (2024), we employ an Oracle routing scheme as a performance upper bound. This scheme selects the expert exhibiting optimal performance on a given evaluation dataset, thus representing a non-zero-shot approach.

## 5.3 MAIN RESULTS

Table 1 presents the comparison results among our GLIDER and six baselines on both held-in and held-out settings. To further illustrate the performance, we also include the results of Oracle Expert, which has extra access to the task identities of expert modules and evaluated datasets and can be regarded as an *upper bound*.

**T0 Setting.** In the T0 task set, the following observations can be drawn: ❶ For the held-in tasks, *i.e.* T0-HI, GLIDER significantly outperforms other baselines and almost matches the performance of Oracle Expert upper bound. ❷ For T0-HO and BBL tasks, GLIDER achieves the best performance among all the methods, including Oracle Expert upper bound. ❸ GLIDER has negligible lower performance, *i.e.* 0.01%, compared to the Expert Merging baseline in BBH but outperforms it by around 12% on T0-HO and 1.5% on BBL. Besides Expert Merging, GLIDER outperforms all other methods on BBH, including the Oracle Expert upper bound.

## 5.4 ABLATION STUDY AND FURTHER INVESTIGATION

**Ablation on the global routing scale $\alpha$.** To illustrate how the specialization and generalization abilities change as we scale the coefficient $\alpha$ of the global routing score, we conduct the ablation study of $\alpha$ ranging $\{1, 3, 10, 100, 1000, 3000\}$. As shown in Table 2, we present experimental results of the T0 task set on both held-in and held-out tasks. For held-in tasks, *i.e.* T0-HI, GLIDER can select the optimal $\alpha$ to scale the global routing score. For held-out tasks, *i.e.* {T0-HO, BBH, BBL}, GLIDER produce either the optimal $\alpha$ (for BBH) or the sub-optimal $\alpha$ with slightly lower performance to the optimal ones (for T0-HO and BBL).

Table 1: Performance evaluated on the T0 set and FLAN set. We present the performance on both held-in tasks (*i.e.* T0-HI) and held-out tasks (*i.e.* T0-HO, BBH, and BBL). We compare the following methods: (1) performance upper bound, *i.e.* Oracle Expert; (2) zero-shot baselines, *i.e.* Multi-Task Fine-Tuning, Expert Merging, Arrow, and Phatgoose; (3) few-shot baselines, *i.e.* LoRA Hub and GLIDER. We mark the best performance besides the upper bound (*i.e.*, Oracle Expert) in **bold**.

| Method | T0 | | | | FLAN | |
| | T0-HI | T0-HO | BBH | BBL | BBH | BBL |
| --- | --- | --- | --- | --- | --- | --- |
| Oracle Expert | 69.60 | 51.60 | 34.90 | 36.60 | 38.90 | 45.40 |
| Multi-Task Fine-Tuning | 55.90 | 51.60 | 34.90 | 36.60 | **38.90** | **45.40** |
| Expert Merging | 30.73 | 45.40 | **35.30** | 36.00 | 34.60 | 34.00 |
| Arrow | 39.84 | 55.10 | 33.60 | 34.50 | 30.60 | 29.60 |
| Phatgoose | 61.42 | 56.90 | 34.90 | 37.30 | 35.60 | 35.20 |
| LoRA Hub | 31.90 | 46.85 | 31.35 | 31.18 | 34.50 | 30.54 |
| GLIDER | **68.04** | **57.78** | 35.29 | **37.46** | 35.07 | 35.52 |

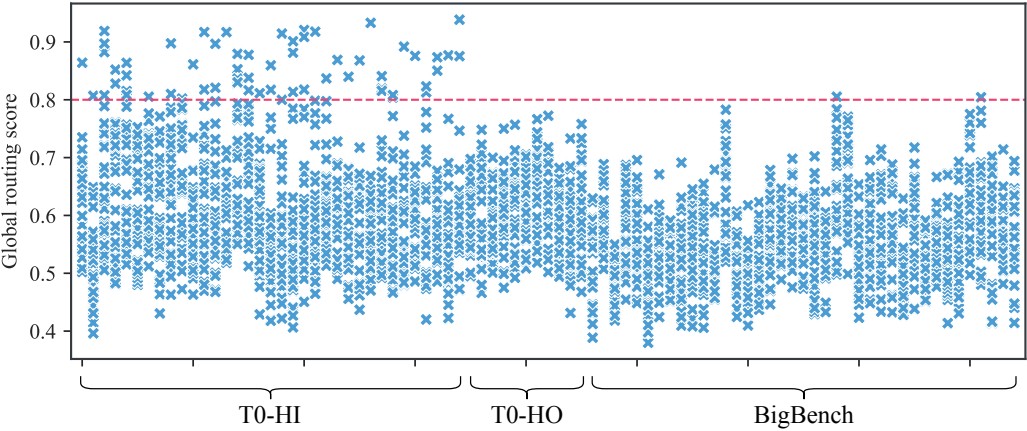

Figure 3: Global routing scores for tasks in the T0 set. The red horizontal line indicates our design threshold of $0.8$. Each column represents an evaluated task from T0-HI, T0-HO, BigBench using T0 held-in experts. All global routing scores for each task are plotted, corresponding to the 35 experts in total.

**Ablation on the routing strategy.** There exists a trade-off between performance and efficiency when using different `top-k` routing strategies (Ramachandran & Le, 2019). To investigate the impact of routing strategy in GLIDER, we evaluate `top-k` routing of k in $\{1, 2, 3\}$. Moreover, we further evaluate the `top-p` routing (Huang et al., 2024c; Zeng et al., 2024) of p in $\{25\%, 50\%, 75\%\}$, where each token selects experts with higher routing probabilities until the cumulative probability exceeds threshold p. As shown in Table 3, we can draw the following conclusions: (1) For `top-k` routing, k = 2 shows comparable or better performance than k = 3, particularly for T0-HO and BBH, while offering improved efficiency. (2) For `top-p` routing, higher p values consistently yield better performance at the cost of efficiency. Therefore, we use `top-2` routing in GLIDER by default.

**Investigation on the threshold design of global scores.** As described in Section 4, we compute the scale $\alpha$ for global scores using the formula $\alpha = \gamma * \mathbb{I}_{\{\max(\mathbf{s}^{\text{glob}}) - 0.8 > 0\}} + \beta$, where we establish a threshold of $0.8$ to differentiate evaluated tasks. Figure 3 presents the global routing scores for each task in the T0 set to motivate the rationale behind this design. For all held-in tasks (*i.e.*, T0-HI), at least one expert (typically the oracle expert trained on the evaluated task) achieves global routing scores exceeding $0.8$. Consequently, GLIDER applies a higher $\alpha = 100$, enabling effective identification of tasks corresponding to a specifically trained expert and enhancing retrieval of this oracle expert. For nearly all held-out tasks (*i.e.*, T0-HO and BigBench), no global routing score surpasses $0.8$, prompting GLIDER to utilize a lower $\alpha = 3$. Two exceptions among the held-out

Table 2: Ablation on the instruction coefficient $\alpha$. We mark the best performance in **bold** and the performance corresponding to the selected $\alpha$ by GLIDER in blue.

| | T0 | | | |
|---|---|---|---|---|
| $\alpha$ | **T0-HI** | **T0-HO** | **BBH** | **BBL** |
| 1 | 62.20 | 57.04 | 35.05 | **37.79** |
| 3 | 63.40 | 57.78 | **35.29** | 37.46 |
| 10 | 65.52 | **57.98** | 34.80 | 37.04 |
| 100 | **68.04** | 53.22 | 31.73 | 34.97 |
| 1000 | 66.88 | 52.91 | 30.71 | 34.31 |
| 3000 | 66.69 | 52.37 | 30.03 | 33.24 |

Table 3: Ablation on the routing strategy. GLIDER employs `top-2` routing. We mark the best performance among `top-k` and `top-p` routing in **bold**, respectively.

| | T0 | | | |
|---|---|---|---|---|
| **Method** | **T0-HI** | **T0-HO** | **BBH** | **BBL** |
| Top-1 | 67.92 | 56.07 | 33.91 | 35.82 |
| Top-2 | 68.04 | **57.78** | **35.39** | 37.46 |
| Top-3 | **68.05** | 57.52 | 35.08 | **38.55** |
| Top-25% | 67.94 | 56.53 | 34.10 | 36.32 |
| Top-50% | 67.91 | 57.25 | 35.07 | 37.49 |
| Top-75% | **68.00** | **57.86** | **35.38** | **38.65** |

tasks are `bbq_lite_json` and `strange_stories` in BigBench, as shown in the figure, where one score marginally exceeds 0.8 in each case. For these two, GLIDER employs the higher $\alpha = 100$, resulting in performance improvements of 1.3% and 2.9% respectively over $\alpha = 3$, thus showing the effectiveness of our design.

## 6 CONCLUSION

This paper introduces GLIDER, a novel multi-scale routing mechanism that incorporates both global semantic and local token-level routers. By leveraging the semantic reasoning capabilities of LLMs for global expert selection and refining these choices with a learned local router, GLIDER addresses the limitations of existing methods that often perform poorly on held-in tasks. Our empirical evaluation on T0 and FLAN benchmarks, using T5-based experts, demonstrates that GLIDER achieves substantial improvements in held-in task performance while maintaining competitive generalization on held-out tasks. These findings suggest that incorporating global semantic task context into routing mechanisms is crucial for building robust and practically useful routing-based systems.

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

# APPENDIX

# A LLM FOR TASK INSTRUCTION GENERATION.

## A.1 PROMPT TEMPLATE

We use the following prompt with 3 randomly selected samples for each task to generate its description. The prompt is then fed into the `gpt-4-turbo` OpenAI API to get the generated task descriptions.

> *The following are three pairs of input-output examples from one task. Generate the task instruction in one sentence that is most possibly used to command a language model to produce them. In the instruction, remember to point out the skill or knowledge required for the task to guide the language model.*
>
> *- Input:*
> *- Output:*
>
> *- Input:*
> *- Output:*
>
> *- Input:*
> *- Output:*

## A.2 EXAMPLES OF THE GENERATED INSTRUCTIONS

We provide several examples of LLM-generated instructions in this section.

**WikiBio** (Lebret et al., 2016) (T0 Held-In):

- *Create a short biography using the provided facts, demonstrating knowledge in historical and biographical writing.*
- *Write a short biography based on the given factual bullet points, demonstrating proficiency in summarizing and transforming structured data into coherent narrative text.*

**CommonGen** (Lin et al., 2020) (T0 Held-In):

- *Generate a coherent sentence using all the given abstract concepts, requiring the skill of concept integration to form a meaningful sentence.*
- *Generate a coherent sentence by creatively combining a given set of abstract concepts.*

**COPA** (Huang et al., 2024b) (T0 Held-Out):

- *Identify the most logically consistent sentence from two given options based on the provided context, demonstrating reasoning and causal relationship skills.*
- *Generate the most likely outcome for a given scenario by choosing between two provided options based on contextual clues and causal reasoning.*

**Date Understanding** (Srivastava et al., 2023) (BigBench-Hard):

- *Calculate the date based on the given information and present it in MM/DD/YYYY format, ensuring that you accurately account for day, month, and year changes.*

**Hindu Mythology Trivia** (Srivastava et al., 2023) (BigBench-Lite):

- *Generate the correct answer by making use of your knowledge in Hindu mythology and culture.*

## B    DEMONSTRATING COMPOSITIONAL GENERATION

In addition to significant improvements on held-in tasks, GLIDER demonstrates strong performance on held-out tasks, showcasing its generalization capability. To further examine this ability to handle unseen tasks by composing experts, we provide specific task examples illustrating the association between selected experts and the evaluated task. As Figure 1 shows, GLIDER primarily selects two experts for the COPA (T0 held-out) task, corresponding to CosmosQA and QuaRel. The following three examples from these tasks demonstrate their close semantic relationship:

- **COPA**:
    - Question: *Everyone in the class turned to stare at the student. Select the most plausible cause: - The student's phone rang. - The student took notes.*
    - Answer: *The student's phone rang.*

- **CosmosQA:**
    - Question: *That idea still weirds me out . I made a blanket for the baby 's older sister before she was born but I completely spaced that this one was on the way , caught up in my own dramas and whatnot . Luckily , I had started a few rows in white just to learn a stitch ages ago , and continuing that stitch will make an acceptable woobie , I think . According to the above context, choose the best option to answer the following question. Question: What did I make for the baby . Options: A. I made a carseat . B. None of the above choices . C. I made a crb . D. I finished a pair of booties .*
    - Answer: *D.*

- **QuaRel:**
    - Question: *Here's a short story: A piece of thread is much thinner than a tree so it is (A) less strong (B) more strong. What is the most sensical answer between "Thread" and "Tree"?*
    - Answer: *Thread.*

