# OpenReview forum: "Glider: Global and Local Instruction-Driven Expert Router"
_ICLR.cc/2025/Conference — ICLR 2025 Conference Withdrawn Submission_

### Official Review · Reviewer_kpST · 2024-10-20

**Soundness:** 2
**Presentation:** 3
**Contribution:** 2
**Rating:** 3
**Confidence:** 3

**Summary:**

The authors propose GLIDER, which is a token-level and module-level hierarchical routing strategy for combining pools of parameter-efficient finetuned expert models, incorporating both local and global information over the specialization of the expert models. The local component learns a gating between LoRA modules which selects the best layer module from the pool of experts for each token. The global component uses an LLM to encode the overall task expertise of each expert, which is then incorporated into the routing scheme to enhance the routing such that it is sensitive both to local module-wise expertise and overall global expertise of the aggregated models. This scheme hopes to maintain strong generalizability to unseen tasks without sacrificing performance on held-in tasks.

**Strengths:**

1)
The core idea -- that incorporating global information of the specialization of finetuned expert models into local routing schemes can improve expert aggregation algorithms -- is intuitive and persuasive.


2)
The use of an LLM to encode global semantic information of the overall expert specialization is a creative method for effectively integrating the required global context

**Weaknesses:**

1)
My first concern relates to the overall problem setting of the paper and its core motivation of improving performance on held-in tasks. The authors claim that existing MoErging methods often prioritize generalization to unseen tasks at the expense of performance on held-in tasks, and indeed in Table 1 the authors report as one of their main results that GLIDER significantly outperforms baselines on held-in tasks. However, performance on held-in tasks is deemed unimportant precisely because we already have access to the specialized expert trained on that exact held-in task, and so we can always retain performance on any given held-in task by simply using the expert specialized to that task. Indeed, this is the reason why Phatgoose does not report results for held-in tasks. For this reason, the 'Oracle Expert' recorded in Table 1 for held-in tasks is not an Oracle but an attainable result for which we always have access - it is just the expert specialized to that given task.

So in this sense, given that for held-in tasks GLIDER still underperforms the expert specialized to that given task, I'm not yet convinced of one the proposed main benefits of GLIDER, since for any given held-in task we could easily get better performance by just selecting the corresponding expert specialized to that task.

Fundamentally, I'm not yet persuaded that the performance gains on held-in tasks justify the claim that GLIDER is an overall superior model, since by the problem setting these are not tasks we need to optimize for. If the authors can provide justification for why performance on held-in tasks is indeed important and why selecting GLIDER over the corresponding specialized expert for a given held-in task would be preferable, then I would be happy to change my score, but as it stands I'm concerned that a large proportion of GLIDER's performance gains are on tasks that we need not optimize for.

2)
A second concern is that GLIDER's architecture, by the authors' own acknowledgement, is basically identical to Phatgoose for the local component of the hierarchical router. This being the case, the contribution of the paper is more so appending a global context component to Phatgoose, rather than an entirely new model. It would therefore be informative to consider alternative backbones for the local component of the router, for example Arrow. This could help isolate the contribution of the global routing component and help to demonstrate the robustness of potential improvements brought about by the proposed inclusion of global information.

3)
Some grammar / spelling related issues:

Lines 53-54: 'However, MoE methods that train experts from scratch while MoErging utilizes  a decentralized...' -> delete 'that'

Line 72: 'retrieve the correct expert for all token at every module' -> tokens should be plural

Line 264: 'our goal is to build a MoErging method that dynamically routing queries' -> should be 'routes queries'

Line 286: 'this work specifically focuses in LoRA' -> focuses 'on' LoRA

Lines 288-289 'Given the $t^{th}$ input token activation $u_i$ -> should be $u_t$ I'm assuming?

Line 332: 'added before the model to independently process the fully query' -> process the 'full' query

Line 348: 'the output of the module for token activation $u_i$ is computed as $Wu_i + \sum_{k \in \xi_{top}} w_k * B_kA_ku_i$ -> It looks like you've forgotten to actually define $w_k$, I'm assuming it's the softmaxed affinity score, but you've left it undefined.

**Questions:**

My first question relates to the core motivation of improving held-in performance and the necessity of doing so given that for any held-in task, we always have access to the expert specialized to that task. Could the authors explain scenarios in which using GLIDER is preferable to simply selecting the specialized expert for a known held-in task?

My second question is to what extent is GLIDER more of a novel component to local routing schemes that aims to encode global context, as opposed to an entirely new model. If GLIDER is more a novel component than an entire model, then I think the authors should include ablation studies on the local router choice, in particular using Arrow.

---

> ### Comment · Reviewer_kpST · 2024-11-26
> **Any response from the authors?**
>
> If the authors have any aspects of the review they'd like to address then let me know.
>
> Looking forward to your responses.

---

### Official Review · Reviewer_rzEk · 2024-10-31

**Soundness:** 2
**Presentation:** 2
**Contribution:** 1
**Rating:** 3
**Confidence:** 4

**Summary:**

the paper studies ensembles of LLM models (model MoErging), proposing a technique for selecting experts to route tokens to at global (for selection of experts for in profile tasks)  and local levels (to have more flexibility to handle out of distribution tasks)

the paper is incremental in nature (with small differences with respect to Phatgoose, from architectural design choice, to experimental settings and way too many details, to the point it feels it should be named Phatgoose++ instead of Glider)

although appealing, the approach is heurisitic in nature: given this, one would have expected a significantly larger experimental part, including a wider range of tasks (and possible comparison points beyond those adopted in Phatgoose), and a statistical relevant comparison of improvements

**Strengths:**

this work introduces a routing mechanism to tradeoff between local and global experts, to increase performance on held in tasks, without compromising capability to handle held out tasks.

the goal is clear and the approach is simple (as it is heuristic in nature).

**Weaknesses:**

# evaluation
as this work has no theoretical basis,  one would have expected a significantly larger experimental part to convince the reader of the generality of the approach on
- a significant wider range of tasks (and possible comparison points beyond those adopted in Phatgoose),
- further exhibiting a statistical relevant comparison of improvements

this is not the case, so the paper execution is far from being convincing.

Additionally, while the main advantage of this work is to increase performance of held-in tasks, authors additionally point out advantages that are too thin to be worth noting; and they do so in a disturbingly biased manner.  For instance, authors claim 0.9% over held out tasks over Phatgoose (in bold), but the 0.9% (actually 0.88%) is the maximum observed across 5 held out datasets in Tab 3 (for the other 4 is 0.39%, 0.16% -0.53% and 0.3%)

# empirical evidence of global router

the work is motivated to find *semantical* resemblance beyond tasks. however, the approach on held out tasks seems to leverage *syntactical* resemblance . Fig  1 shows held out tasks to systematically select two experts (one of which seems to be further common to a couple of tasks).  Yet appendix B just shows the tasks to be syntactically similar: i..e, the Q&A pair has a cloze format, which is rather typical of  simple benchmarks. As such, I am little reassured that the performance generalization will be maintained on more complex tasks, and this work is far from fully elucidating the robustness of the proposed method.

Quantitative assessment of global experts over a wider range  of diverse tasks  (say few tens of datasets per type of answer) would have allowed to get true insights about the nature of global experts (e.g., whether the identified expert triggers "cloze" type answers, irrespectively of the semantic of the question)

**Questions:**

# incremental addition of new experts

One question that is not addressed is the incremental addition of new experts/datasets pair, and what are the consequences (on normalization, etc.). It seems it should be trivial to do this incrementally, but a discussion in the appendix (and an example use-case when, say a previously held-out task becomes held-in so that the router now uses global instead of local weights) would certainly improve the quality of the paper

# limit of global router

the global router is constructed using a sample of 3 questions, which may be ok for very simple and low-diversity datasets, but not for more complex  and diverse tasks,. a more in-depth study of global router across a wider range of datasets with quantitative assessment of global router policies seems necessary

---

> ### Comment · Reviewer_rzEk · 2024-11-26
> **rebuttal?**
>
> I haven't seen any author rebuttal ?

---

### Official Review · Reviewer_S6dB · 2024-11-02

**Soundness:** 3
**Presentation:** 3
**Contribution:** 2
**Rating:** 5
**Confidence:** 3

**Summary:**

This paper presents GLIDER, a method that combines global semantic and local token-level routing for LLMs. The key innovation is using an LLM to generate task instructions that guide expert model selection globally, while using token-level routing locally. Tested on T5-based models with T0 and FLAN benchmarks, GLIDER improves performance on held-in tasks by 6.6% while maintaining strong generalization on held-out tasks. This shows that incorporating LLM-driven semantic understanding helps achieve better routing compared to pure token-level approaches.

**Strengths:**

* The paper leverages LLMs to generate semantic task descriptions, providing global context for routing decisions which is a unique approach not explored in previous routing methods
* The paper well address the limitations of current approaches (focusing on either held-in or held-out tasks) and provides a novel solution integrating both.

**Weaknesses:**

* The experiments focus solely on T5, an older encoder-decoder architecture. The effectiveness of GLIDER on modern decoder-only models (like GPT family, LLaMA, etc.) remains unproven, which is crucial given these are now the mainstream architectures for LLMs.
* Table 1 lacks clarity on evaluation metrics and methodological details. Without clear metric definitions and evaluation protocols, it's difficult to fully assess and compare the reported improvements.
* The routing design will bring extra computational overhead, how will GLIDER's inference latency change compare with the normal LoRA decoding methods (vLLM's lora inference module).

**Questions:**

* Could you explain why T5 was chosen as the primary architecture for evaluation? Have you conducted any preliminary experiments with decoder-only models e.g. (LLama3-8B etc)?
* Could you provide more details about the evaluation metrics used in Table 1? What exactly do these numbers represent?
* Since this design integrates routing into the LoRA inference procedure, could you provide a detailed analysis of the additional computational overhead? How much will the inference latency be affected by such design?

---

### Official Review · Reviewer_yu6t · 2024-11-04

**Soundness:** 2
**Presentation:** 1
**Contribution:** 2
**Rating:** 5
**Confidence:** 3

**Summary:**

This paper focuses on addressing the trade-off between performance improvement and generalization ability in expert modules. It assumes that this issue arises from the lack of global semantic context in token-level routing strategies, it then seeks to resolve this problem by combining global semantics with token-level information through the use of both global and local expert routers during routing.

**Strengths:**

1. The insights of the paper are to be praised.
2. Very interesting topic and focus on the router optimization.
3. Use the big model and small model together to solve the problem
4. Experimental results are good.

**Weaknesses:**

1. Writing and Presentation:The paper could benefit from some polishing. There are a number of typos and semantic issues, and the overall formatting could be improved for better readability. Additionally, some figures are a bit challenging to interpret. For instance, Figure 1 is only referenced in Appendix B but appears as the first figure in the Related Works section, which can disrupt the flow and clarity for the reader.

2. Clarity of Background and Concepts: The background and explanation of key concepts in the paper could be clearer. While there are many references to ideas and works by Yadaav, the connections and explanations aren’t sufficiently detailed, which may leave readers a bit confused. In my initial reading, I found myself questioning whether the discussion pertained to a Mixture of Experts (MoE) scenario or Model Merging.

3. Logical Flow and Mathematical Details: The paper seems to lack some logical coherence, especially regarding mathematical descriptions and derivations. There are no thorough mathematical proofs provided, and the modeling of the scenario feels a bit scattered across different sections. Some variable explanations are incomplete, which can be frustrating. Moreover, discussions around problems and solutions could be more precise. For instance, when mentioning that routing strategy issues stem from a lack of global semantics, it would be helpful to have more rigorous mathematical reasoning or experimental evidence to support this claim.

4. Inconsistencies: There are some notable inconsistencies in the paper. For example, in line 85, it mentions that the Global Router selects the top-2 experts based on global semantics. However, the description of the Global Router algorithm starting from line 329 doesn’t reference any top-2 (or top-k) selection process. The top-k expert selection is only brought up later around line 347, based on the final score calculated from the weighted sum of global and local affinity scores. Clarifying these points would enhance the overall coherence of the paper.

5. The experimental design could use some improvement. The main experiments lack detailed explanations, and Figure 3 is somewhat unclear. Many of the experimental configurations seem to mirror those from Muqeeth's work without providing enough context, which might raise questions about the originality of this study. Additionally, the ablation experiments focus on relatively trivial variables, while more significant factors—such as the differences between excluding and including the global semantics generated by GPT-4-turbo—are overlooked. Addressing these points could enhance the depth and rigor of the research.

**Questions:**

Inovation and contribution:
1. The core of this paper combines the scores from the Global Router and the Local Router, then performs a top-k operation based on the combined scores. a. Line 85: The descriptions of the algorithm in the preceding and following contexts are clearly conflicting, making it contradictory to describe the algorithm effectively. b. Relying on GPT-4 turbo to gather information from the full text means the process heavily depends on GPT-4's capabilities. Furthermore, as seen in line 346 and the subsequent ablation experiments, the value of $\alpha$ is significant, indicating a high weight assigned to the Global score.
2. The paper extensively references the work of Muqeeth but fails to explain the rationale behind these references, raising concerns about the originality of the article.
3. Relying on GPT-4-turbo for global semantic information raises some questions about the overall workload and originality of the research. From the formula in line 346 and the subsequent ablation experiments, it appears that the global score heavily influences the final affinity score, making the expert selection largely dependent on GPT-4-turbo. Additionally, the potential latency issues introduced by using GPT-4-turbo during inference are not addressed, which is a significant concern. Overall, placing so much emphasis on GPT in the paper could weaken its persuasive impact.

Format and Typo Issues:
1. 60) Formatting issue: The left parenthesis is missing.
2. 67-68) The sentences are semantically repetitive.
3. 140) What does the arrow ($\rightarrow$) signify here?
4. 216) This figure is drawn in a hard model; it’s unclear what it is trying to convey.
5. 279) The explanation of $\Psi$ is missing.
6. 289-290) The notation used here is confsued.
7. 301) There is no explanation for the newly introduced $\sigma$.
8. 319) Suddenly introducing the variable 𝑡 t from line 289 is confusing.
9. 323) Why isn’t the normalization process presented in a formula? Section 4.2 is poorly written, relying solely on descriptive language, lacking organization.
10. 354) The T0 held-in dataset is omitted.

---

### Author Response · Authors · 2024-11-26
**General Response**

Dear Reviewers & ACs & SACs & PCs,

Thank you for taking the time to review our submission and for providing your valuable feedback. We deeply appreciate your thoughtful comments and insights, which have helped us better understand the strengths and weaknesses of our work.

After careful consideration, we have decided to withdraw the paper from ICLR this year. We intend to use your feedback to improve the paper further and hope to resubmit it to a future venue once the necessary revisions have been made.

Thank you once again for your time and effort in reviewing our work.

\
Warmest regards,

Authors

---

### Note · Authors · 2024-11-26

I have read and agree with the venue's withdrawal policy on behalf of myself and my co-authors.